# Comparison of Vaccination Regimens on Immune Responses Using PED Replicon Vaccine: A Field Trial in PED-Negative and PED-Positive Thai Swine Farms

**DOI:** 10.3390/ani15020273

**Published:** 2025-01-19

**Authors:** Chaitawat Sirisereewan, Thanh Che Nguyen, Nanthiya Iampraphat, Hongyao Lin, Leonardo Ellerma, Pisit Sirithanyakul, Roongtham Kedkovid, Roongroje Thanawongnuwech

**Affiliations:** 1Department of Veterinary Pathology, Faculty of Veterinary Science, Chulalongkorn University, Bangkok 10330, Thailand; cs.chaitawat@gmail.com (C.S.); chethanh.nguyen.pdr@gmail.com (T.C.N.); 2Center of Excellence for Emerging and Re-emerging Infectious Diseases in Animals and One Health Research Cluster, Faculty of Veterinary Science, Chulalongkorn University, Bangkok 10330, Thailand; 3Veterinary Diagnostic Laboratory, Faculty of Veterinary Science, Chulalongkorn University, Bangkok 10330, Thailand; nanthiya.i@chula.ac.th; 4MSD Animal Health Innovation Pte Ltd., Perahu Road, Singapore 718847, Singapore; hongyao.lin@msd.com; 5MSD Animal Health (Phils) Inc., 8767 Paseo De Roxas, 26/F AIA Tower, Makati 1200, Philippines; leonardo.ellerma@merck.com; 6Intervet Thailand Ltd., 999/9 The Offices at Central World, 37th Rama 1 Road, Pathumwan, Bangkok 10330, Thailand; pisit.sirithanyakul@merck.com; 7Department of Veterinary Medicine, Faculty of Veterinary Science, Chulalongkorn University, Bangkok 10330, Thailand; 8Centre of Excellence in Swine Reproduction, Chulalongkorn University, Bangkok 10330, Thailand

**Keywords:** porcine epidemic diarrhea, porcine epidemic diarrhea virus, replicon vaccine, pig, vaccination

## Abstract

This study explored the use of a new replicon vaccine against porcine epidemic diarrhea in different vaccination protocols. The findings demonstrate how different vaccination strategies influence antibody production, emphasizing the need to select appropriate regimens to optimize herd immunity. Vaccination with the replicon vaccine during acclimatization showed a promising trend in increasing IgA and IgG antibody levels. Notably, using the replicon vaccine as a booster after a modified-live virus vaccination led to superior neutralizing antibody responses. Additionally, combining a replicon vaccine booster during gestation with a killed vaccine during acclimatization effectively enhanced maternal-derived immunity. These results highlight the potential of the replicon vaccine as a tool to mitigate the economic losses of porcine epidemic diarrhea infection.

## 1. Introduction

Porcine epidemic diarrhea virus (PEDV) is an enteric swine pathogen causing porcine epidemic diarrhea (PED), leading to substantial economic losses to the global swine industry. It is a large, enveloped RNA virus classified under the genus *Alphacoronavirus* within the *Coronaviridae* family. While PEDV can infect pigs of all ages, the highest mortality occurs in neonatal piglets under ten days old. Infected pigs suffer from severe gastrointestinal illness due to the degeneration and necrosis of intestinal epithelial cells and villi, impairing absorption and causing symptoms like vomiting, diarrhea, and death [1,2]. These events have led to severe economic losses, with the U.S. pork industry alone losing approximately USD 1 billion during the PED outbreak [3]. In Thailand, PEDV was first reported in 2007 [4] and is now considered endemic [5]. To control the disease in endemic areas, strict biosecurity measures and effective husbandry practices are implemented to induce passive lactogenic immunity against the virus.

Since the first outbreak of African swine fever virus (ASFV) in Asia, increasingly stringent biosecurity measures implemented after the ASFV outbreak have been associated with a significant reduction in the spread of many pathogens among herds including PEDV [6,7,8]. However, PEDV outbreaks have been reported in several countries including China and Thailand, despite the increased biosecurity [9,10]. Hence, apart from biosecurity, inducing passive lactogenic immunity to PEDV is crucial for protecting neonatal piglets. To establish passive lactogenic immunity, the immunization of sows with modified live PEDV vaccines or exposure to the virus through feedback has remained a routine practice to induce lactogenic immunity, which is transferred to piglets via maternal colostra and milk [11,12,13,14]. However, both approaches have limitations. Gut feedback, which involves exposing the breeding herd to infected materials or intestines, often results in inconsistent immune responses and carries the risk of introducing additional pathogens [14,15]. Interestingly, recurrent PED outbreaks in Korean swine farms have been associated with the use of feedback material [16]. Additionally, gut feedback plays a significant role in accelerating the virus evolution rate [11] and carries a risk of contamination by other pathogens [14]. Modified live vaccines (MLVs) have been associated with safety risks as there is a risk of genetic recombination or reversion to virulence through interactions with wild-type strains [17,18]. The use of live virus in both feedback and MLV has raised concerns about its long-term survival in the environment and the potential for asymptomatic gilts to spread PEDV to farrowing barns, posing major challenges for eradication efforts [19]. While inactivated vaccines are safer, the immune response generated is less comprehensive and shorter-lasting compared to that induced by live attenuated vaccines [20]. Inactivated vaccines have also demonstrated varied efficacy in piglets born to vaccinated sows, depending on whether sows are pre-exposed or naïve to PED [1,21], with better immunity observed in pre-exposed sows.

A new vaccine platform using alphavirus RNA particle-based technology has demonstrated the ability to induce strong cellular and humoral immune responses in various animal models of infectious diseases [22]. In this system, the foreign gene of interest is inserted in place of Venezuelan equine encephalitis virus (VEEV) structural genes, generating a self-amplifying RNA replicon capable of expressing the gene of interest upon introduction into the cells. The self-amplifying RNA replicon directs the translation of large amounts of protein in transfected cells, reaching levels as high as 15–20% of the total cell protein [23]. As the RNA replicon does not contain any of the VEEV structural genes, the RNA is propagation-defective. The RNA replicon can be packaged into replicon particles (RPs) by supplying the VEEV structural genes in trans in the form of promoter-less capsid and glycoprotein helper RNAs. When the helper RNAs and RNA replicon are combined and co-transfected into the cells, the RNA replicon is efficiently packaged into single-cycle, propagation-defective RP, which is then used in the vaccine formulation [24].

The replicon forms the basis of the Sequivity^®^ RNA particle vaccine platform (Merck Animal Health, Rahway, NJ, USA), which is currently licensed in the U.S. for multiple swine applications. Protection has been demonstrated in clinical settings for several diseases such as COVID-19 [25], swine influenza [26], and PED [27].

In summary, boosting lactogenic immunity through vaccination is crucial in endemic areas as an aid to mitigate production losses associated with PED. The study aimed to investigate various vaccination protocols using a newly available PED replicon vaccine (PED-RP) and assess their impact on responses following the acclimatization phase and/or farrowing.

## 2. Materials and Methods

### 2.1. Experimental Design

To understand the immune responses induced by the PED-RP in replacement gilts during acclimatization, two experiments were conducted on farms with differing PED status and distinct time points. Experiment 1 involved a PED-negative herd with no history of PEDV outbreaks and continuing active management to maintain the PEDV-free status. This farm utilized a killed PED vaccine as a preventive measure to mitigate losses in the event of an outbreak. Experiment 2 was conducted on a PED-positive herd with a history of ongoing PED outbreaks. PEDV field strains were able to be isolated from infected piglets during these outbreaks. Immune response parameters were collected post-vaccination in a similar manner, with minor procedural differences. All animal protocols were approved by the Chulalongkorn University Animal Care and Use Committee (IACUC No. 2031029).

#### 2.1.1. Experiment 1

In this experiment, 120 replacement gilts were randomly assigned to four groups, each receiving a distinct vaccination protocol. Vaccinations included either the killed PED vaccine already in use as part of the routine farm vaccination protocol, the PED-RP (Sequivity^®^ PED, Merck Animal Health, Rahway, NJ, USA), or a combination of the two. Vaccinations were administered during the acclimatization phase. For selected groups, the PED-RP was also given as a booster dose during the gestation phases (four weeks before farrowing). Both vaccines were administered intramuscularly. A summary of vaccination protocols is provided in Table 1. IgA and IgG antibody responses were evaluated in the serum and colostrum at the end of the acclimatization phase and at farrowing.

#### 2.1.2. Experiment 2

In this experiment, 42 replacement gilts were randomly assigned to two groups, each following a specific vaccination protocol during the acclimatization phase. Group A (MLV/MLV; *n* = 21) received two doses of the PED MLV vaccine, while Group B (MLV/PED-RP; *n* = 21) received one dose of the PED MLV vaccine followed by one dose of PED-RP. Both vaccines were administered intramuscularly. IgA antibody levels and serum neutralization titers were evaluated at the end of the acclimatization period. Due to farm-specific constraints, this experiment was limited to the acclimatization phase.

### 2.2. Sample Collection

Blood samples were collected from all groups at two time points: on the initial day of acclimatization (before vaccination) and at the end of the acclimatization phase (after vaccination) in both experiments. The samples were centrifuged at 2500 rpm for 5 min at room temperature to collect the sera, which were subsequently stored at −80 °C until use.

In Experiment 1, the colostrum samples were aseptically collected from sows within 12–24 h postpartum. A volume of 3–5 mL per sow was obtained and stored at −20 °C until further analysis.

### 2.3. Evaluation of PEDV-Specific IgA and IgG in Serum and Colostrum, and Serum Neutralization

In Experiment 1, all serum and colostrum samples were sent to the MSD Animal Health Center for Diagnostic Solutions (Boxmeer, The Netherlands) for PEDV-specific IgA and IgG antibody detection using in-house ELISA tests. Samples were considered anti-PEDV IgA-positive if the optical density (OD) value was ≥0.2, and anti-PEDV IgG-positive if the OD value was ≥0.3.

In Experiment 2, the IgA antibody levels were measured using the IDEXX PEDV IgA Ab test kit (IDEXX Laboratories, Westbrook, ME, USA) following the manufacturer’s instructions. Briefly, the test kit validity was confirmed if the mean OD values for the positive and negative controls were 0.55 and 0.10, respectively. A sample was considered positive if its sample-to-positive (S/P) ratio was ≥0.50. The S/P ratio was calculated using the formula: [(sample OD − mean OD negative control)/mean OD positive control − mean OD negative control)]. Serum neutralization (SN) tests were performed at the end of acclimatization following a previously reported protocol [28] with minor modifications. Briefly, Vero cells were cultured in a medium comprising equal parts minimum essential medium (MEM) and Dulbecco’s modified Eagle medium (DMEM) supplemented with 10% fetal bovine serum, 0.3% tryptose phosphate broth, 0.02% yeast extract, and 10 μg/mL trypsin. Serum samples were subjected to twofold serial dilutions starting at 1:2. A field strain of PEDV at 100 TCID50/50 μL was mixed with an equal volume of the diluted serum, and the mixture was incubated for 1 h at 37 °C with 5% CO_2_. Following incubation, the mixture was transferred to a 96-well plate containing Vero cells. Cytopathic effects (CPEs) were monitored daily, and SN titers were determined after 48 h.

### 2.4. Statistical Analysis

Data were analyzed using analysis of variance (ANOVA) followed by Tukey’s multiple comparison tests, or the Kruskal–Wallis test followed by Dunn’s multiple comparison tests, as appropriate. All statistical analyses were performed using GraphPad Prism version 8 for Windows (GraphPad Software Incorporated, San Diego, CA, USA).

## 3. Results

### 3.1. Evaluation of Immune Responses in Vaccinated Gilts’ Sera During the Acclimatization Period

The effectiveness of various vaccination regimens on immune responses during acclimatization was investigated in both PED-negative and PED-positive farm conditions.

Experiment 1: The serum samples were analyzed for the presence of IgA and IgG before and after vaccination during the acclimatization phase. The levels of anti-PEDV IgA and IgG in all groups are shown in Figure 1 and Figure 2, respectively. Briefly, the anti-PEDV IgA remained low across all groups throughout the study, in contrast to the IgG results. All groups exhibited increased mean serum IgG levels by the end of the acclimatization phase (post-vaccination period). Notably, pigs in Group B and Group C, which received two doses of the PED-RP, had significantly higher mean serum IgG levels compared to Group A and Group D (*p* < 0.05), which were immunized with two doses of the killed vaccine. The percentages of IgG seropositive gilts were 100% across all vaccination regimens.

Experiment 2: In this experiment, the serum IgA levels were evaluated in gilts vaccinated with two protocols, MLV/PED-RP and MLV/MLV. The results showed that 23.8% (5/21) of gilts vaccinated with the MLV/PED-RP protocol were seropositive, while no seropositive cases were found in the MLV/MLV group. However, the mean IgA levels in both groups remained below the cut-off threshold (Figure 3). In addition, SN titers increased in both vaccination regimens. Gilts boosted with PED-RP demonstrated significantly higher SN titers compared to those vaccinated with the MLV/MLV protocol (Figure 4).

### 3.2. Evaluation of Immune Responses in Vaccinated Gilts’ Colostrum at Postpartum Period

To assess maternal-derived immunity, colostrum samples were collected within 24 h postpartum from pigs in Experiment 1. Anti-PEDV IgA was detected in the colostrum of 33% (3/9) of pigs in Group A, 33% (3/9) in Group B, 56% (5/9) in Group C, and 100% (5/5) in Group D. Notably, pigs in Group D exhibited significantly higher PEDV-specific colostrum IgA levels compared to Group A (*p* < 0.05). The mean IgA levels for each group are shown in Figure 5. For colostrum IgG detection, all pigs tested positive for anti-PEDV IgG antibodies. However, Group C demonstrated significantly higher mean IgG levels compared with Group A (*p* < 0.05). The mean IgG levels across the groups are presented in Figure 6.

## 4. Discussion

PEDV has been recognized as one of the most economically devastating swine viruses, causing high morbidity and mortality in neonatal piglets. To establish herd immunity, the immunization of sows with PED vaccines or exposure to the virus through feedback has remained a routine practice to induce lactogenic immunity, which is transferred to piglets via maternal colostrum and milk [11,13,14,19]. However, both approaches have limitations such as inconsistently inducing lactogenic immunity or having biosecurity risks, respectively. Furthermore, commercially available PED vaccines offering broad cross protection against diverse field strains are not available yet, and different vaccine regimens have shown varying protective outcomes against PED [29]. Therefore, tailoring vaccine strategies based on available PED vaccines may enhance lactogenic immunity more effectively.

The PED status of a swine farm is a critical factor in selecting the appropriate type of PED vaccine. On PED-negative farms, noninfectious vaccines such as killed vaccines, DNA vaccines, and PED-RP are preferred to minimize the risk of introducing live viruses onto the farm. Conversely, PED-positive farms may benefit from strategies incorporating both MLV vaccines and gut feedback to enhance immunity. In this study, we investigated the effectiveness of various vaccine regimens under two distinct farm conditions: a combination of killed PED vaccine and PED-RP were tested in a PED-negative farm setting (Experiment 1), while a combination of PED MLV vaccine and PED-RP was assessed in a PED-positive farm (Experiment 2).

During acclimatization, the results from both experiments showed that immunized replacement gilts did not exhibit mean PEDV IgA antibody levels in serum above the cut-off value. Interestingly, priming with the MLV vaccine followed by PED-RP showed a promising trend, achieving 23.8% seropositivity in replacement gilts and significantly enhancing the neutralizing antibody titers compared to two doses of the MLV vaccine. Our results are consistent with previous reports that found that two doses of the killed PED vaccine administered during the acclimatization phase did not produce sufficient serum IgA antibody levels, with only 50% seropositivity observed [10]. However, replacement gilts receiving two doses of PED-RP elicited higher IgG antibody levels in serum than that of the pigs immunized with two doses of the killed PED vaccine. Unfortunately, the IgG antibody levels were not assessed in replacement gilts immunized with combinations of the MLV vaccine and the PED-RP, warranting further investigation in future studies. Our findings indicate that vaccination during the acclimatization phase alone may not be sufficient to significantly enhance the PEDV-IgA antibody levels, although a trend of increasing IgA levels was observed after vaccination in all vaccinated groups. Based on our results, we postulate that additional immunization during the gestation phase may be required to achieve optimal lactogenic immunity.

In Experiment 1, administering an additional dose of PED-RP during gestation resulted in significantly higher mean colostrum IgA levels compared to the groups that received only two vaccine doses during the acclimatization phase. This finding underscores the importance of a gestational booster, particularly in PED-negative herds, as it significantly enhances the colostrum IgA levels. Previous studies have shown that PED vaccination during the second trimester effectively stimulates the gut–mammary gland–secretory IgA axis, providing robust lactogenic immune protection against PEDV in piglets [12]. Additionally, serum IgA has been identified as having the strongest correlation with IgA antibodies and neutralizing antibodies present in colostrum [30], highlighting its pivotal role in the protective immune response transferred to piglets. Furthermore, a booster dose of PED-RP during gestation, combined with two doses of PED-RP during acclimatization, also elicited higher IgG antibody levels in the colostrum. However, due to farm constraints, the IgA and IgG in colostrum in Experiment 2 were not tested.

Based on our results, vaccination with PED-RP during acclimatization shows a promising trend in increasing the IgA and IgG antibody levels. Interestingly, a booster dose of PED-RP following MLV vaccination elicited superior neutralizing antibody responses. Similarly, a booster dose of PED-RP during gestation, combined with a killed vaccine during acclimatization, effectively enhanced the lactogenic immunity compared to other regimens. This robust immune response is likely due to the RNA replicon’s ability to express the target antigen at high levels [31,32,33]. A single self-amplifying RNA can produce up to 200,000 copies of RNA, significantly boosting protein expression [32]. Therefore, incorporating PED-RP as a booster dose can elicit a stronger immune response and can be effectively implemented in both PED-positive and PED-negative farms.

## 5. Conclusions

In conclusion, different vaccination strategies induced varying levels of antibody production, underscoring the importance of selecting appropriate regimens to optimize herd immunity and protect piglets against PED infection. Booster vaccination during gestation with PED-RP, a novel platform vaccine, significantly increased the colostrum IgA levels, demonstrating its potential as an effective booster. This study highlights the versatility of PED-RP in various vaccination protocols, providing valuable insights for its implementation in both PED-positive and PED-negative farms. Further research on combining PED-RP with MLV or feedback strategies in PEDV-endemic farms is needed to refine the vaccination protocols and enhance maternal-derived immunity.

## Figures and Tables

**Figure 1 animals-15-00273-f001:**
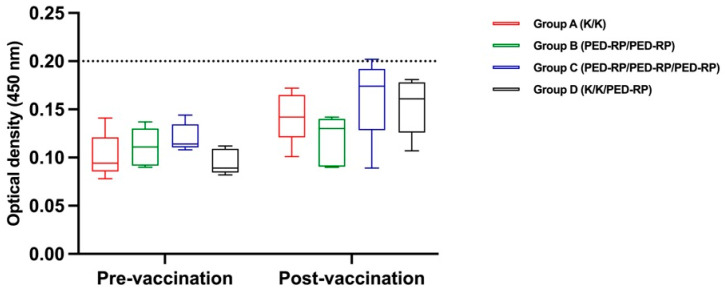
Mean anti-PEDV IgA antibody levels in serum from vaccinated gilts were compared among the study groups at the end of acclimatization. Dashed lines indicate the cut-off value of 0.2 for IgA. One-way ANOVA or the nonparametric Kruskal–Wallis test was used to analyze the differences between groups.

**Figure 2 animals-15-00273-f002:**
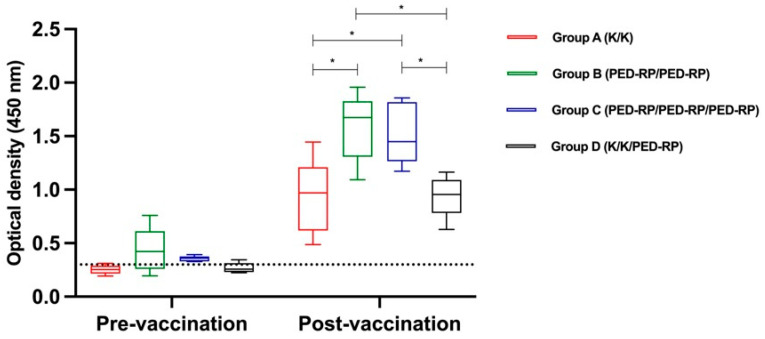
Mean anti-PEDV IgG antibody levels in serum from vaccinated gilts were compared among the study groups at the end of acclimatization. Dashed lines indicate the cut-off value of 0.3 for IgG. One-way ANOVA or the nonparametric Kruskal–Wallis test was used to analyze the differences between groups. Asterisks (*) indicate statistically significant differences between the groups.

**Figure 3 animals-15-00273-f003:**
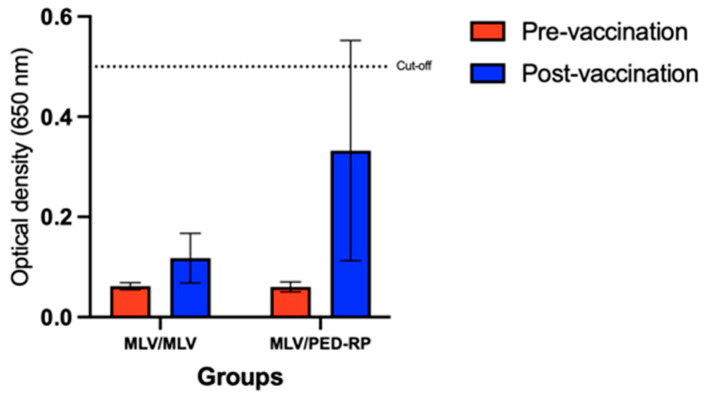
Mean serum anti-PEDV IgA antibody levels in the vaccinated gilts across different vaccine regimens at the end of acclimatization in Experiment 2. Dashed lines indicate the cut-off value of 0.5 for IgA. One-way ANOVA or the nonparametric Kruskal–Wallis test was used to analyze the differences between groups.

**Figure 4 animals-15-00273-f004:**
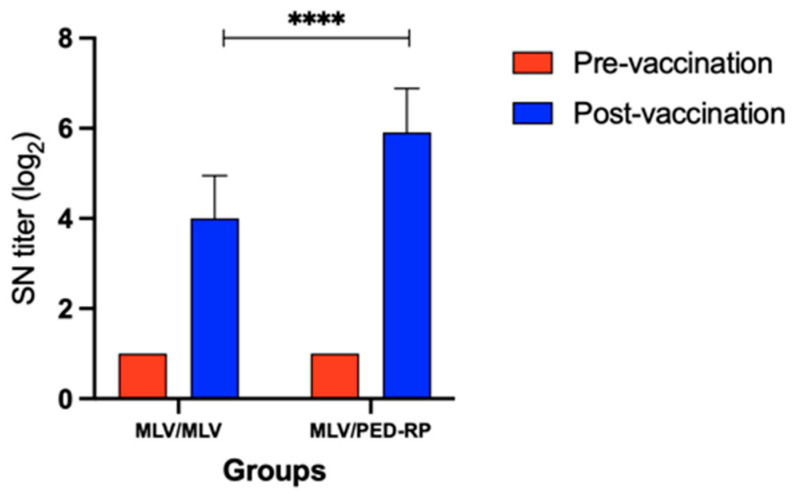
Neutralizing antibody titers across different vaccine regimens at the end of acclimatization in Experiment 2. One-way ANOVA or the nonparametric Kruskal–Wallis test was used to analyze the differences between groups. Asterisks (****) indicate statistically significant differences between the groups.

**Figure 5 animals-15-00273-f005:**
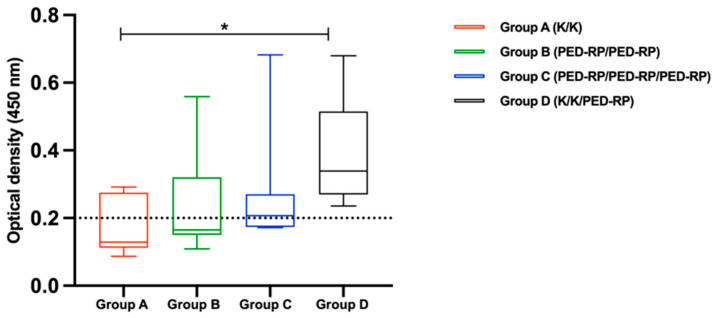
Mean anti-PEDV IgA antibody levels in colostrum collected from vaccinated gilts within 24 h after parturition were compared among the study groups. Dashed lines indicate the cut-off value of 0.2 for IgA. One-way ANOVA or the nonparametric Kruskal–Wallis test was used to analyze the differences between groups. Asterisks (*) indicate statistically significant differences between the groups.

**Figure 6 animals-15-00273-f006:**
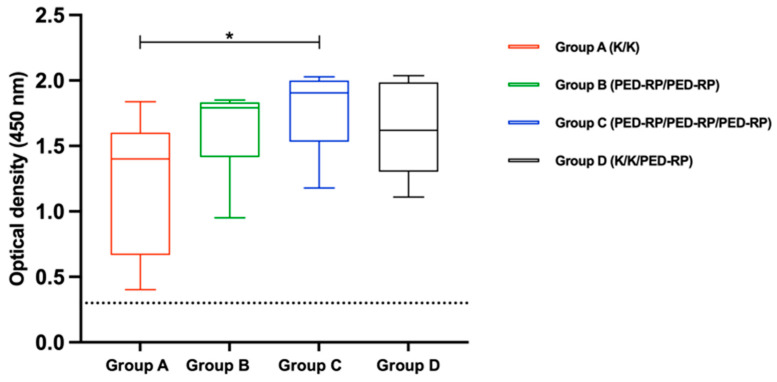
Mean anti-PEDV IgG antibody levels in colostrum collected from vaccinated gilts within 24 h after parturition were compared among the study groups. Dashed lines indicate the cut-off value of 0.3 for IgG. One-way ANOVA or the nonparametric Kruskal–Wallis test was used to analyze the differences between groups. Asterisks (*) indicate statistically significant differences between the groups.

**Table 1 animals-15-00273-t001:** The details of the experimental groups and vaccination protocols used in this study.

Experimental Groups (*n* = 120)	Vaccination Protocols ^1^	Doses of Vaccination	
		Acclimatization Phase	Gestation Phase ^2^
Group A (*n* = 30)	K/K	K, 2 shots	-
Group B (*n* = 30)	PED-RP/PED-RP	PED-RP, 2 shots	-
Group C (*n* = 30)	PED-RP/PED-RP/PED-RP	PED-RP, 2 shots	PED-RP, 1 shot
Group D (*n* = 30)	K/K/PED-RP	K, 2 shots	PED-RP, 1 shot

^1^ K = killed vaccine for porcine epidemic diarrhea; PED-RP = replicon vaccine for porcine epidemic diarrhea; ^2^ Four weeks before farrowing.

## Data Availability

The data presented in this study are available on request from the corresponding author.

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
