# Peer review of "Comparison of Vaccination Regimens on Immune Responses Using PED Replicon Vaccine: A Field Trial in PED-Negative and PED-Positive Thai Swine Farms"

_animals, 2025, doi:10.3390/ani15020273_

Round 1
Reviewer 1 Report
Comments and Suggestions for Authors
This study explores the potential of a PED replicon vaccine with several vaccination protocols. It was found that the replicon vaccine as a booster after a modified-live virus vaccination led to strong neutralizing antibody responses. Additionally, combining a replicon vaccine booster during gestation of gilts with a killed vaccine during acclimatization effectively enhanced colostrum IgA production which are crucial for piglets protection. The study also illustrated how different vaccination strategies influence antibody production, emphasizing the need for selecting appropriate regimens to optimize herd immunity. There are several questions for the authors to clarify:
1. In the experiment 1, there was no significant difference of serum IgA level between groups. How was there a significant difference of colostrum IgA level between Group A and D? Shouldn't the IgA in the colostrum come from sow blood?
2. The morbidity and mortality rates of piglets in different groups were not provided. The article didn't tell whether the vaccination protocols help prevent the diarrhea disease in the pig groups.
3. It is not proper to mention the names of the companies of the vaccines, as different results of the experiments may influence their commercial impact, which may cause unnecessary conflict.
Reviewer 2 Report
Comments and Suggestions for Authors
The authors present an interesting article on the impact of PED vaccination strategy and antibody titers in serum and colostrum in swine. The authors identified that a killed/killed/PED replicon strategy induced the highest colostrum antibody titer which is important for mitigation of PED. Overall, the manuscript is interesting and brief. The authors state farm restrictions for the duration of the experiment. A challenge experiment of the neonates would be a crucial next step, but is beyond the scope of the current paper. Determining vaccine durability would also be an interesting finding. Could a sow provide protection into the next breeding cycle?
The authors conclusions are in-line with the data presented. Clearly stating that serum titers did not meet positive criteria for figure 1 would enhance the veracity of the claims.
The difference in neutralizing antibody and overall titer is interesting. Is there an established correlate of protection of any of the PED vaccines? If so, how do these results compare? The methods for the neutralization need more explanation, what were the minor modifications?
Minor points:
Line 33 and with similar statements, if numbers are available, please use them. How much per year does PED cost the swine industry? "substantial" is vague.
Figure 5 and 6, restate the time of collection in the legend. Since there is only 1 collection point, please consider consolidating group names/strategies under the figure legend in place of "Experimental Groups".
Reviewer 3 Report
Comments and Suggestions for Authors
Comments on animals-3384999
The present study compared the immune responses induced by different vaccination protocols containing various vaccines against porcine epidemic diarrhea (PED) in Thai swine farms. The work is clinically relevant. However, the study was not well-designed. Several concerns should be addressed.
Major concerns:
1. The neutralizing antibody titers were not tested in Experiment 1 and the maternally derived immunity was not assessed in Experiment 2. Why?
2. The authors are suggested to compare the protection of the immunized sows or their offspring from virulent challenge conferred by the different vaccination protocols.
3. PED-negative and PED-positive are confusing. The authors should specify the status correctly, PEDV-positive or PED-seroconverted?
4. The manuscript should be revised by Native English speakers.
Comments on the Quality of English LanguageThe English writing should be improved.
Round 2
Reviewer 3 Report
Comments and Suggestions for Authors
The manuscript should be revised.
For more details, please refer to the attachment.

The English writing should be improved.
